# Application of Elastin-like Polypeptide in Tumor Therapy

**DOI:** 10.3390/cancers14153683

**Published:** 2022-07-28

**Authors:** Xianggang Shi, Dongfeng Chen, Guodong Liu, Hailing Zhang, Xiaoyan Wang, Zhi Wu, Yan Wu, Feng Yu, Qinggang Xu

**Affiliations:** 1School of Life Sciences, Jiangsu University, Zhenjiang 212013, China; 2222018057@stmail.ujs.edu.cn (X.S.); dongfeng@ujs.edu.cn (D.C.); wuyan@ujs.edu.cn (Y.W.); 2Department of Gastroenterology, The Affiliated Suqian First People’s Hospital of Nanjing Medical University, Suqian 223800, China; l11488890@163.com (G.L.); zhl051022@yeah.net (H.Z.); sq80908192@126.com (X.W.); 3Jiangsu Key Laboratory of High-Tech Research and Development of Veterinary Biopharmaceuticals, Jiangsu Agri-Animal Husbandry Vocational College, Taizhou 225306, China; 2007020048@jsahvc.edu.cn

**Keywords:** elastin-like peptides, peptide drugs, temperature response, photosensitizer, tumor therapy

## Abstract

**Simple Summary:**

Elastin-like Polypeptide (ELP) are widely applied in protein purification, drug delivery, tissue engineering, and even tumor therapy. Recent studies show that usage of ELP has made great progress in combination with peptide drugs or antibody drugs. The combination of ELP and photosensitizer in cancer therapy or imaging has made more progress and needs to be summarized. In this review, we summarize the specific application of ELP in cancer therapy, especially the latest developments in the combined use of ELP with photosensitizers. We seek to provide the most recent understanding of ELP and its new application in combination with Photosensitizer.

**Abstract:**

Elastin-like polypeptides (ELPs) are stimulus-responsive artificially designed proteins synthesized from the core amino acid sequence of human tropoelastin. ELPs have good biocompatibility and biodegradability and do not systemically induce adverse immune responses, making them a suitable module for drug delivery. Design strategies can equip ELPs with the ability to respond to changes in temperature and pH or the capacity to self-assemble into nanoparticles. These unique tunable biophysicochemical properties make ELPs among the most widely studied biopolymers employed in protein purification, drug delivery, tissue engineering and even in tumor therapy. As a module for drug delivery and as a carrier to target tumor cells, the combination of ELPs with therapeutic drugs, antibodies and photo-oxidation molecules has been shown to result in improved pharmacokinetic properties (prolonged half-life, drug targeting, cell penetration and controlled release) while restricting the cytotoxicity of the drug to a confined infected site. In this review, we summarize the latest developments in the application methods of ELP employed in tumor therapy, with a focus on its conjugation with peptide drugs, antibodies and photosensitizers.

## 1. Introduction

Several decades ago, Urry et al. [1] isolated and characterized a polypeptide polymer that was previously known to form the resilient component of arterial walls and ligaments, in addition to providing elasticity. In aqueous solution, this polypeptide exhibits a unique phase-transition property and thermally sensitive behavior whereby it coacervates and settles from solution when the temperature reaches its critical limit, resolubilizing at a temperature below the critical limit. This polypeptide protein known as tropoelastin, a non-crosslinked precursor of elastin, is capable of forming ligaments upon coacervation. The core amino acid sequence was found to be composed of valine, proline and glycine arranged as (VPGVG)n, where n is the number of repetitive sequences of the precursor unit [2]. Due to its elasticity the fact that its primary core sequence is derived from human tropoelastin, a precursor of elastin, this biosynthetic polypeptide was named elastin-like polypeptide (ELP). Subsequently, other polypeptides, including collagen and resilin, exhibiting similar stimulus-responsive behavior, such as coacervation and solubilization, following changes in external factors were identified and characterized [3]. Since then, elastin-like polypeptides, resilin-like polypeptides and collagen-like polypeptides have attracted increasing research interest for use in various scientific and biomedical applications [4,5]. The intrinsic properties of these polypeptides allow them to be fine-tuned into different targeted structures in the form of nanoparticles, nanofibers and other forms of nanocomposites suitable for emerging novel applications, among which tissue engineering, drug delivery, bioimaging, biosensors and protein purification are the most commonly investigated [6,7,8,9]. Because the precursor units of these polypeptides (especially ELPs) are largely of human origin, their systemic tolerance properties have been exploited for use as modules for sustained drug delivery in localized anatomical regions of the body and for the biodistribution of injectable proteins without provoking adverse autoimmune responses [10,11].

Following the discovery of its amino acid sequence and an understanding of its stimulus-responsiveness, the pentapeptide repeat units of ELPs were identified and modified to be composed of VPGXaaG (Xaa is a guest residue and can be substituted for any amino acid except for Pro); replacing the guest residue with a specific amino acid can change the physicochemical and biological properties of the ELP [12]. If the guest residue Xaa is Pro, the conformation of the ELP chain is disrupted, which destroys the principal characteristic and useful feature of the biopolymer, i.e., its inverse thermal transition property [13]. Custom design of an ELP core sequence with specific amino acids and sequence length with subsequent specialized modifications and functionalization can help to precisely control its environmental reactivity, mechanical properties and cellular metabolic pathways [14,15,16,17].

Based on different application scenarios, ELPs can be endowed with specific suitable properties for targeted application. The biological and scientific significance of ELPs revolves around their stimulus-responsiveness to temperature, pH and light. With respect to temperature, ELPs exhibit lower critical solution temperature (LCST) behavior whereby the ELP either solubilizes or coacervates in aqueous solutions in response to changes in temperature below or above the critical transition limit [18,19]. Through precise adjustment of the ELP sequence composition, order of amino acids and molecular weight (MW; the number of pentapeptide repeat units), the phase transition temperature (Tt) can be significantly altered to suit the intended purpose. That is, although the basic intrinsic properties of ELPs satisfy the LCST principle, when the temperature is higher than the Tt, the hydrophobic regions of ELPs shed water, aggregate and become insoluble in solution, and when the temperature is lower than the Tt, the ELPs resolubilize [20]. This unique thermally responsive property of ELPs coupled with inverse transition cycling (ITC) has been exploited for the purification of recombinant proteins with considerable success without resorting to conventional chromatographic purification, achieving improved efficiency, quality, yield and recovery of the target protein [21]. The number of pentapeptide repeat units commonly ranges from 10 to 330 and can be accurately adjusted to achieve an optimal Tt of ELPs—usually between 0 °C and 100 °C [14,22,23]. As an intrinsic parameter to precisely control the Tt, the MW of ELPs is inversely related to the Tt, i.e., increasing the MW results in a decrease in the Tt. Moreover, the type of guest residue Xaa used for the design of ELPs can also impact the Tt. Whereas increasing hydrophilic guest residues (such as Gly) increases the Tt, replacing the guest residue with hydrophobic Xaa (such as Val) decreases the Tt of ELPs [24,25]. In general, ELPs with hydrophobic Xaa have a lower phase transition temperature than ELPs with hydrophilic Xaa [10]. Although the Tt of ELPs may be largely controlled by intrinsic parameters, such as the type of Xaa and the number pentapeptide units (MW) [26,27], pH sensitivity, as an extrinsic factor, is also known to play an important role. When a pH-sensitive acidic or basic amino acid is placed in the position of Xaa, the Tt of the ELP becomes pH-dependent. In general, the Tt and solubility increase when the amino acid in ELPs is cationic or anionic (e.g., histidine under acidic conditions or glutamic acid under basic conditions) [28]. For example, when the valine in position 4 of ELP (VPGVG)n is replaced in one of five pentamers by a glutamic acid residue, the aggregation temperature of the ELP becomes remarkably sensitive to pH, and the Tt shifts from 25 °C at both pH 2 and pH 7, to 25 °C at pH 2 and to 70 °C at pH 7 [29]. In addition to thermal responsiveness and pH sensitivity, ELPs can be fine-tuned and endowed with the ability to effectively respond to light by irradiation. Under excitation at a specific light intensity originating from a laser or another source, ELP response to light can be used as an approach to improve photodynamic therapy (PDT) with increased efficiency [30]. For instance, ELP(VPGXaaG)40 (Xaa = K) conjugated to a derivative of tetraphenylethene (TPE-COOH) via an amide bond has been investigated as a potential bioprobe for cell imaging and was found to exhibit lower cytotoxicity and stronger fluorescence than those of TPE-COOH alone [22]. Aside from its potential in bioprobing and bioimaging, the emerging strategy of combining ELPs and photosensitizers for tumor therapy has made promising progress toward improving drug efficiency while reducing the risk of toxicity. Thus, owing to the numerous advantageous properties of ELPs, such as biocompatibility, biodegradability, low immunogenicity and stimulus-responsiveness, as well as its quintessential applications, including protein purification, drug delivery, controlled drug release, micellar carrier construction, tissue engineering and tumor treatment, investigation into its photosensitive behavior towards tumor therapy is essential. In this review, we summarize the application of ELPs as a delivery module in tumor therapy, especially when combined with photosensitizers and therapeutic peptides.

## 2. Clinical Development of ELP-Based Therapeutic Drugs

In the field of tumor therapy, multiple strategies have been developed to manage or treat tumors depending on the severity of the case. Although more sophisticated strategies are available in the clinical phase, radiation therapy, chemotherapy (‘chemodrug’), surgical removal and immunotherapy are currently widely employed. Although chemotherapy often yields exciting results, its side effects consistently concern patient who may have to undergo the process. Whereas chemodrugs efficiently target cancer cells, normal cells are equally attacked and destroyed in the process [31,32]. To limit this unrestricted attack on normal cells and redirect the drugs to target tumor cells, chemotherapeutic agents are modified by attaching them to secondary polymers and inoculated to a target site by localized deposition. In this case, the cytotoxicity of the drug is largely restricted to the tumor site, and translocation is limited due to the high molecular weight of the polymer to which it is conjugated. As a thermally responsive biopolymer, ELPs have been shown to exhibit a tunable property and can be designed to exhibit a determined biochemical behavior suitable for an expected function. Designed to function as a module for drug delivery, ELPs were conjugated with hydrophobic cancer drugs and aggregated into nanoparticles in the vasculature of tumors under mild hyperthermic conditions [33]. Liu et al. [34] observed that after implanting a conjugate of ELP and therapeutic radionuclide into the tumors of xenograft mice and found that the conjugate drug, while showing a longer residence time, also helped to delay tumor progression when compared to the control. In an attempt to downregulate the expression of ovarian-cancer-promoting gene NT5C3, Ramamurthi et al. [35] designed an ELP-gemcitabine conjugate for aggregation and accumulation at the target site and a controlled release of gemcitabine in a pH-dependent manner for the control of ovarian cancer. The results indicated a significant downregulation of the NT5C3 gene by the ELP-drug conjugate compared with the control and when the drug was used alone. The cytotoxic nature of the conjugate drug may have improved pharmacokinetics, enhanced cell penetration and accumulation, as well as decreased clearance. Moreover, several studies have reported successful design of a complex of ELP-drug/peptide conjugates, such as ELPs conjugated with doxorubicin for efficient tumor or cancer targeting and drug delivery, [36,37], ELPs with vascular endothelial growth factor (VEGF) for treatment of preeclampsia [38] and renovascular disease [39], ELPs conjugated with paclitaxel for breast cancer treatment by inhibiting the proliferation of the MCF-7 cell line, stabilizing microtubules structures, arresting cell division at the G2/M stage and inducing apoptosis [40]. Although successive experimental findings from both in vivo and in vitro studies continue to present positive results of ELP-peptide/chemodrug conjugates, it is important to examine the possibility of any adverse effect they may elicit should they be employed in clinical trials.

## 3. ELPs Increase the Pharmacokinetic Properties of Therapeutic Drugs

Proteins have increasingly been recognized as attractive natural therapeutic macromolecules for the treatment of a wide variety of diseases. Although targeted therapeutic proteins/peptides commonly referred to as peptide drugs (PDs) have high specificity and activity toward diseases, a major limitation hindering their clinical application is their short circulating half-life [41,42]. Protein/peptide-based drugs can be categorized into two groups: antibody drugs (such as single-chain antibodies and heavy-chain antibody fragments) and therapeutic peptides. Usually, the small size of PDs (such as antibody drugs) results in increased clearance from circulation, limiting the drugs’ ability to produce the desired immune effect [43]. To extend the circulating half-life of protein-based therapeutics, several site-specific modification studies have been conducted wherein target proteins are either functionalized with large macromolecular compounds or genetically fused with known polypeptides for heterogenous expression, purification and application. The most commonly studied macromolecular polymer used to covalently bind to proteins is polyethylene glycol (PEG) by a process known as PEGylation, which introduces changes to the physicochemical properties of the protein molecule, thereby increasing its molecular size and weight, hydrophilicity and intermolecular interactions, in addition to changes in conformation. These changes introduced by the PEG help to decrease protein clearance while improving delivery to target sites and have been utilized to modify several commercial and clinically available peptide drugs [44,45,46], for which the efficacy and circulating half-life of PEGylated therapeutic proteins have been investigated [44,47]. Similar to PEG, human serum albumin (HSA) has been investigated and clinically tested as a binding module to improve the pharmacokinetics and pharmacodynamics of therapeutic proteins due to its low toxicity, high biocompatibility and apparently stable half-life within the human system [48,49]. Despite the significant advantages and success achieved by introducing PEG and HSA in target proteins, a highly efficient, cost-effective and reliable precision genetically designed protein-polymer fusion complex with high yield, specificity, bioactivity and well sustained pharmacodynamics and pharmacokinetics is required. The emergence of ELPs as a classical binding module to facilitate the fusion and delivery of therapeutic molecules or provide a surface site for the attachment and display of PDs to increase pharmacokinetic properties has become the subject of intense research focus [50,51]. Thus, by using ELPs as a delivery module, target molecular drugs or peptide drugs can be encapsulated, attached or displayed on the surface of ELP nanoparticles. The classical usage of ELPs in disease treatment takes full advantage of their temperature- and pH-responsive characteristics. According to extensive reviews, ELPs are inherently endowed with a tunable Tt property that allows them to self-assemble into nanoparticles (or nanoworms) either by physical, chemical or genetic modification through conjugation with molecular or peptide drugs [17,52,53].

Present studies have indicated that upon fusion with ELPs, the circulating half-life of PDs, including single-chain antibodies (anti-CD20 scFv, anti-CD99 scFv, anti-FLT3 scFv and anti-EGFR heavy-chain antibody) and therapeutic peptides (AP1 and p50), in the body increased significantly while inducing enhanced cytotoxic damage to the target tumor site or cell line [43,54,55]. As shown in a study by Vaikari et al. [55], the average residence time of the bioengineered PD nanoworm α-CD99-ELP examined for its efficacy against leukemia in mice was dramatically increased, with a circulating half-life of 16 h and a mean residence time of 21.3 h and could reside in liver, spleen and kidney tissue for more than 96 h after infiltration. Moreover, the leukemia burden was significantly reduced, with prolonged survival among the α-CD99-ELP-treated group compared with that of the control group. In a similar study, Sarangthem et al. [54] evaluated the antitumor effect of double therapeutic peptide-ELP conjugate by developing a chimeric polypeptide composed of interleukin-4 receptor-targeting peptide (AP1), the proapoptotic peptide (KLAKLAK) and ELPs for targeted treatment of glioblastoma tumors. The AP1-ELP-KLAK-treated group showed a cell death rate of up to 2~3 fold higher than that of the control group, with dramatically reduced tumor growth as a result of inducing apoptosis in tumor-bearing mice. The study further indicated that although AP1 and KLAK are targeting molecules in glioblastoma and ELP improves the half-life of PD, the absence of any of these three peptides in a single treatment results in a diminishing therapeutic effect against glioblastoma, as shown in the group treated with PBS, ELP-KLAK and API-ELP. This effect may have been a result of increased clearance of the peptides as compared with the AP1-ELP-KLAK-treated group, where apoptosis could still be detected after 72–96 h, which indicates enhanced half-life. Whereas improving the half-life and residence time is among the reasons for ELPs usage as a drug delivery module, their eventual degradation and clearance from the system are essential due to their non-biological significance. However, it must be noted that after the drug has been released and its task is achieved, like all other drugs, it must be cleared from the system. As such, clearance mechanisms, such as exposure to circulatory proteases, altering the temperature of the location where the ELPs were deposited and coagulated to lower the Tt for ELP solubilization or by introducing a biodegradable linker in the case of localized hydrogel-forming photosensitive ELP-peptide-photosensitizer conjugate and the generation of oxidants to disrupt disulfide bridges within the hydrogel, are among the processes that may facilitate the metabolism, sera and renal clearance of used-up ELP modules [27,56]. In summary, whereas fusion of proteins and peptide drugs with ELPs eliminates limitations such as faster circulation clearance, offering a prolonged half-life and residence time, the conjugate drug can also be fine-tuned to target specificity and high cytotoxic potential. However, in the absence of antibody receptor molecules, the therapeutic efficacy of the constructed ELP conjugate is significantly reduced.

## 4. ELPs Fused with Specific Peptides to Improve Antitumor Efficacy of Drugs

Aside from enhancing the circulating half-life of PDs, the second significant effect of ELPs on PDs is their ability to increase the accumulation of targeted drugs at the tumor site in a temperature-dependent manner. The most classic application of this strategy is fusion of ELPs with cell-penetrating peptides (CPPs) or with a specific peptide that targets tumor antigens (TPs) in combination with antibiotics or small-molecule drugs (Dox, PTX). ELP-CPP or ELP-TP fusion protein complexes can improve cell penetration and drug targeting and prolong the half-life and reduce the toxicity of small-molecule drugs to normal cells and tissues. Compared to free drugs, ELP-CPP drug modules show significant cell penetration and improved cancer-cell-killing efficacy [57]. Moreover, a combined ELP-TP drug module has been reported to show a more efficient tumor-cell-targeting specificity compared to free drugs [58,59]. Thus, the product formed by the fusion of two or more tumor-targeting drugs, with the inclusion of ELPs as a delivery or carrier module, can significantly improve drug distribution, homing to target site, cell penetration and pharmacokinetics, in addition to demonstrating effective tumor cell cytotoxicity [40,57,58,59,60]. Whereas ELP-CPP/TP-drug conjugates have improved targeting ability, ELPs alone or in combination with a drug (ELP-drug conjugate) do not show unique binding or specificity for tumors, probably due to the lack of antigen-antibody specificity. For example, as previously indicated, Sarangthem et al. [54] suggested that whereas AP1-ELP-KLAK conjugate had significantly enhanced intratumoral localization, prolonged retention time and significantly inhibited tumor growth, ELP-KLAK without IL-4 receptor targeting (AP1) failed to induce apoptosis and exhibited limited localization and retention at the tumor site. In a hyperthermic therapy scenario, at an ELP phase transition temperature of 41~42 °C, fusion peptide drugs or encapsulated drugs can passively accumulate and aggregate at the tumor site or on the tumor cell surface [37,61]. Improved cell penetration and tissue accumulation were observed for the treatment of glioblastoma after a dominant negative Notch inhibitory peptide (dnMAML) and ELP conjugate modified with cell penetrating peptide SynB1 were synthesized. The enhanced cell penetration ability SynB1-ELP-dnMAML conjugate was found to induce apoptosis and cell cycle arrest by inhibiting the growth of D54 and U251 glioblastoma cell lines under hyperthermic conditions [62,63]. Similarly, to inhibit the growth of mammary cancers, NFkB inhibiting peptide p50 conjugated with SynB1 and ELP also exhibited improved tumor penetration while inducing apoptosis, inhibiting breast cancer cell growth and blocking the intranuclear import of NFkB [64]. These studies clearly indicate that drug accumulation at tumor sites and on tumor cell surfaces significantly increases internalization and residence time when fused with ELPs under optimal conditions [62,64]. Moreover, as shown by Thomas and Dragojevic [64], the concentration of ELP-fused therapeutic drugs detected in cells of the hyperthermia-treated group was ~1.5 times higher than that of the non-hyperthermia-treated group. The corresponding increase in drug efficacy among the hyperthermia group compared to the non-hyperthermia group indicates targeted accumulation of the therapeutic drug at the tumor cell sites. The above studies showed that peptide drugs fused with ELPs can have improved pharmacokinetic properties, especially when the drugs are largely accumulated at the treatment site, where they can be easily internalized into cells. As observed by Ryu et al. [63], intravenously delivered ELP-CPP/TP conjugates are quickly and easily cleared from the system under physiological conditions but are accumulated at tumor sites when the temperature is increased to 42 °C, indicating that ELP-CPP-TP conjugate accumulation occurs only under hyperthermic conditions. This results in prolonged average residence time after uptake and accumulation according to both in vitro and in vivo studies. Taken together, fusion of ELPs with specific peptides not only improves drug penetration, delivery and homing to targeting sites but also prolongs the half-life of the drug. ELPs also increase the accumulation and effective concentration of the target drug at the pathological site, in addition to reducing non-specific diffusion of the drug and reducing toxicity (Table 1).

## 5. Conjugation of ELPs and Peptide/Antibody Drugs with Photosensitizers

Photodynamic therapy (PDT) is a clinically recognized non-invasive treatment approach that can be used to treat series of cancers and non-malignant diseases by triggering a series of cell death mechanisms [69,70,71]. Generally, PDT relies on three basic ingredients: photosensitizer (PS), oxygen and light [72]. Photosensitizers used for PDT are specialized photosensitive molecules that localize to a target cell or tissue and transfer energy from light to an oxygen molecule to generate reactive oxygen species (ROS). The reactions that occur in PDT are restricted to the immediate locale of the light-absorbing photosensitizer, indicating that all biological responses emanating from this reactive process are only activated in specific areas of the tissue exposed to light-limiting collateral damage to healthy, unaffected regions [73]. The main mechanisms by which PDT mediates tumor destruction have been described as occurring in three main pathways [73]. (1) The ROS generated from the reaction between photosensitizers and light can kill tumor cells directly. However, complete tumor eradication is often not achieved due to non-homogenous distribution of photosensitizers within the tumor. (2) Tumor infarction induced by vasculature shutdown has been studied in some PDT procedures. The formation of new blood vessels to facilitate the supply of nutrients and oxygen is the main lifeline of tumor growth and sustenance. However, limiting these processes and shutting down oxygen and nutrient supply through the destruction of blood vessels significantly delays tumor growth. (3) The induction of immune response at PDT-treated sites is another observed mechanism of tumor destruction. Infiltration of immune response cells and biomarkers at PDT-treated tumor sites indicates the activation of immune response, which allows for the augmentation of T cells, lymphocytes and macrophages to be accumulated, followed by the release of cytotoxic cytokines to speed up tumor death [69,74].

Taking advantage of the phase-transition property of ELPs, their amino acid sequence, composition and molecular weight can be fine-tuned to either solubilize or coacervate according to the conditions of the tumor microenvironment. Because tumors are highly vascularized, ELPs can take advantage of this medium to be transported and deposited at target sites in order to achieve highly homogenous tissue infiltration while limiting their clearance rate. Thus, combining ELPs with PSs endows the PSs with enhanced tissue permeation and increased half-life without affecting their ability to respond to light and to generate ROS [75]. For more efficient drug delivery and sustained specific activity over a prolonged period, researchers have evaluated the incorporation of photosensitizers, ELPs and peptide/antibody drugs into a single conjugate and examined their pharmacokinetic and pharmacodynamic behaviors (See Figure 1).

For example, according to a report by Mukerji et al. [76], the photosensitizer chlorine e6 (Ce6) was chemically conjugated to a cysteine-functionalized ELP by N-terminal carboxyl-amine conjugation in the presence of dicyclohexylcarbodiimide and N-hydroxysuccinimide. Ce6, which is activated by near-infrared light at 660 nm, was used to generate a high-yield singlet oxygen (^1^O_2_), which induced the formation of disulfide bridges between the cysteine to form a stable hydrogel both in vitro and in vivo. The stable hydrogel formed by cysteine crosslinking served as a reliable therapeutic depot for a sustained release of ROS, resulting in significant tumor growth inhibition when compared with either Ce6 or cELP alone. (See Figure 2 for an illustration of the formation of disulfide bridges by photoirradiation).

Similarly, Phill et al. [43] decorated the nanoparticles of ELP-modified Illama heavy-chain antibody fragment (VHH), targeting the epidermal growth factor receptor (EGFR) overexpressed in various cancers with the photosensitizer IRDye 700 by conjugation and determined its cell-killing efficiency. The study indicated that the PS-ELP-VHH conjugate/micelle demonstrated tumor-cell-killing efficiency in a light-dependent manner, indicating an efficient binding between the antibody (VHH) and the antigen (EGFR), as well as a sustained release of ROS generated by the light and the PS in the presence of oxygen. Ibrahimova et al. [30] also examined the photo-oxidative response of ELP conjugates for application in PDT. In their study, TT1, a peripherally substituted carboxy-Zn(II)-phthalocyanine derivative, was selected as photosensitizer (PS), acting as a photo-oxidation catalyst at around 680nm. The TT1 was conjugated to the N terminal of the ELPs modified with an alkyne group [77]. Under specific photoirradiation conditions, the hydrophobic TT1-ELP[M1V3-40] (large micellar) turned into an amphiphilic TT1-ELP[M(O)1V3-40] and self-assembled into small micelles at physiological body temperature, which diffuse more easily into dense tumor surroundings and permit a second photoirradiation, which subsequently leads to more efficient PDT [30]. In another study by Sun et al. [78], a cysteine-rich ELP was genetically fused with human high-mobility group protein 2 (F3) chemoselectively conjugated with polypyrrole (PPy) nanoparticles, after which doxorubicin (DOX) was physically adsorbed onto the PPy-ELP-F3 nanoparticle. The multifunctional non-histone chromosome-binding F3 protein, which has binding specificity for nucleolin, expressed on the membrane of tumor cells can be internalized into the targeted cells and translocated to the nucleus upon binding to the nucleolin. Under laser irradiation, DOX/PPy-ELP-F3 showed enhanced cytotoxicity, which was attributed to the synergistic photothermal and chemical effect of the DOX/PPy-ELP-F3. The ease of cellular penetration offered by F3, the solubility of ELP nanoparticle formation by PPy and the photo-oxidation effect of DOX all contributed to specific nucleus homing and the generation of ROS required to initiate cell death (Figure 3). Narayan et al. [79] reported the synthesis of engineered photo-responsive silk-elastin-like polypeptide polymerized with the monomeric photoreceptor C-terminal adenosylcobalamin binding domain (CarHC) using SpyTag-SpyCatcher chemistry. The resulting hydrogel after photostimulation showed a significantly improved encapsulation and controlled-release performance for L929 murine fibroblast cells in 3D culture, suggesting a potential use of the CarHC-sELP conjugate hydrogel as a potential carrier for the controlled delivery of tumor-killing therapeutic agents. In summary, although it has been shown that photodynamic therapy using a sensitizer as a lone treatment material is effective, conjugation with ELPs and tumor-specific, site-directed target antibodies offers profound advantages, including improved solubility and target homing, effective infiltration and binding, crosslinking at the target site to ensure prolonged residence duration and the controlled generation of cytotoxic ROS required for apoptosis initiation within tumor cells.

## 6. Limitations and Future Perspectives for ELP-Drug Conjugates

Although the combination of surgery and chemotherapy remains a reliable go-to choice in many tumor management cases, the search for an improved therapeutic strategy with little to no life-threatening effect continues to be pursued. Hence, improving the therapeutic potency of polypeptide-mediated and ELP-based drug delivery systems could represent the next-generation advancement in healthcare in terms of improving the wellbeing of patients. Nonetheless, due to present limitations of such a therapeutic approach, prospective studies could consider addressing some identified issues to improve the efficacy of this polymer-based drug delivery system. Tumors are highly susceptible to extravasation, which accounts for most cases of metastasis and is also the cause of loss of inoculated drug conjugates. Although conjugation of ELPs with photosensitizers may induce crosslinking of ELPs to form a stable hydrogel, nanoparticles or micelles to improve their residence time, the pore size of the vasculature at the tumor site and the size of the ELP-transformed micelles, nanoparticles or hydrogel requires investigation to establish the formation of stable conjugates in tumors resistant to extravasation in the event of fluid leakage. Aside from leakage, it has been demonstrated by multiple studies, including those by Moktan and Raucher [80] and Sarangthem et al. [54], that ELPs have a limited effect on target tissue infiltration in the absence of cell-penetrating peptides. As such, cells treated with ELPs and cytotoxic peptides with CPPs do not have the same tumor-inhibiting effect when compared with groups treated with combined ELP-peptide-CPP conjugates. However, it appears that ELP-peptide drug infiltration and internalization may be cell- or tissue-specific. Iglesias and Koria [81] reported that KLAK-ELP and keratinocyte growth factor (KGF)-ELP can form nanoparticles and be internalized via micropinocytosis through an interaction of the ELP domain with cell-surface heparin sulfate proteoglycans in lung cancer cells.

It is now established that ELP sequence, chain length, guest residue, protein concentration, salt type and concentration and solution pH are all important factors that affect the Tt of ELPs [82]. Hence, when ELPs are fused with other proteins (e.g., CPPs or peptide drugs), the ELP-based fusion proteins may demonstrate a change in Tt [20]. Charged amino acids from non-ELP proteins in a fused ELP-peptide polymer conjugate have the most significant effect on the Tt of ELP-based fusion proteins [20,83]. Therefore, these effects of non-ELP protein residues should be considered when designing ELP-fused protein drugs in order to achieve effective infiltration and coagulation for maximum cytotoxic impact. Although it has been confirmed that hyperthermic conditions are required for localized ELP aggregation, it should be noted that temperature is not the only external factor that can influence aggregation. Increasing ELP MW, altering the guest residue with an amino acid more susceptible to aggregation and changing pH and solute are among the changes that can be introduced to influence ELP coacervation at physiological temperatures. Notwithstanding, it must be noted that temperature-responsive ELPs can serve as a drug reservoir at the tumor site, effectively increasing the drug concentration at the pathological site upon aggregation and reducing the concentration of drugs at non-pathological sites. This process has been examined by modifying ELP-peptide drug conjugates with cell-penetrating peptides, as reported for SynB1-ELP-DOX and SynB1-ELP1-dnMAML, which specifically targeted and infiltrated infected cells [62,63], prolonged the half-life of the drug and had a sustained release effect (such as SynB1-ELP-DOX) [40]. As has been shown, the cell penetration and aggregation of ELP-peptide drug conjugates is effective largely under hyperthermic conditions.

Moreover, combination of ELPs and photosensitizers provides route to develop optically responsive ELPs. At present, light-responsive hydrogels prepared by the combination of ELPs and photosensitizers have the advantages of long-term structural stability and no adverse immune responses in animals [84]. Reversible light-responsive SELP-CarHC hydrogels are another solution in terms of cell encapsulation and release [79]. Although photodynamic therapy is reported to be a highly efficient strategy for treatment of tumors, the use of visible light hinders deep-tissue penetration to target internal tumors, largely restricting the application of this process to external tumors. Self-excitation polypeptides that use an internal light source for excitation were recently investigated [85]. Thus, genetically encoded peptides with chemiluminescence, bioluminescence and Cerenkov radiation potentials in the presence of an internal light source can be used as a photosensitizer and conjugated with ELPs for efficient tumor infiltration.

## 7. Conclusions

In general, ELPs show low to no toxicity but improve the pharmacokinetic properties of the drug to which they are conjugated, indicating that ELPs as a biomaterial have merits and high potential as a module for drug delivery but require improvement to be accepted and utilized for the purposes of drug delivery without the fear of inducing inflammation or adverse immune responses. Combining drugs with ELPs could not only significantly improve the therapeutic effect of conventional drugs but also increase targeting specificity and reduce toxicity to normal cells or tissues.

## Figures and Tables

**Figure 1 cancers-14-03683-f001:**
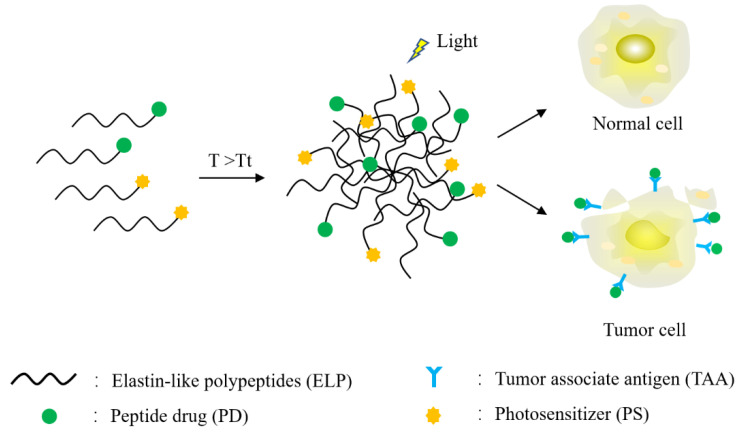
Schematic representation of photodynamic therapy using ELP-PS in combination with ELP-PD. T > Tt indicates that ELP aggregation occurs when the temperature is higher than the Tt; when stimulated by light, the aggregated ELPs release the drug.

**Figure 2 cancers-14-03683-f002:**
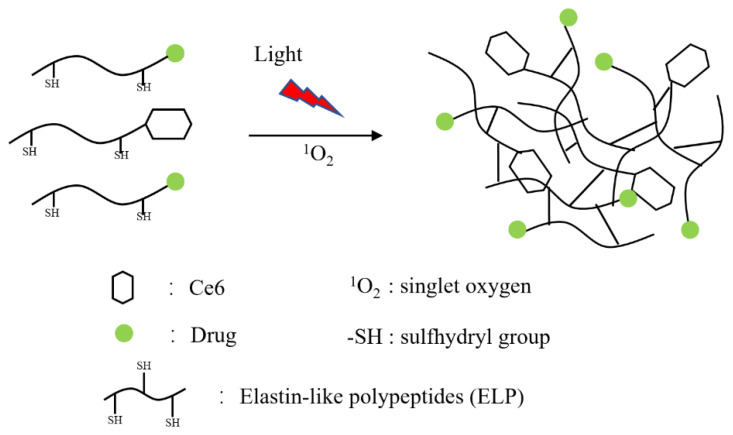
Ce6-ELP crosslinked through disulfide bonds under near-infrared wavelengths.

**Figure 3 cancers-14-03683-f003:**
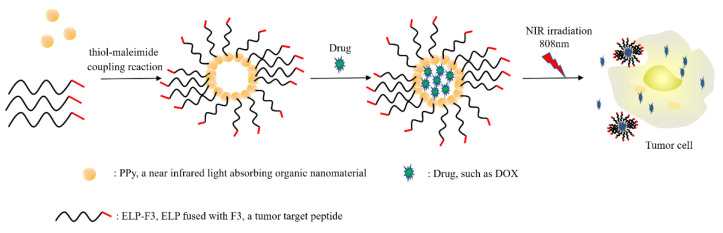
Photosensitizer-assisted photothermal and chemical synergistic cancer therapy.

**Table 1 cancers-14-03683-t001:** Fusion methods of ELPs with specific peptides.

Specific Peptides	Peptides or Molecular Drugs	Functionalization	References
anti-CD20 scFV/anti-FLT3 scFV/anti-CD99 scFV	CD20, FLT3, CD99	Targeting, delivery, safety	[65,66,67]
SynB1/CPP	dnMAML, p50	Penetration, delivery, safety, accumulation, persistence	[62,63,64]
AP1	KLAK	Targeting, accumulation	[54]
SynB1/CPP/mmpL	Dox	Penetration, delivery, reduced toxicity, fixed-point release, accumulation	[40,57,58]
EGF/SynB1	PTX/dnMAML and PTX	Penetration, targeting, reservoir, accumulation, controlled release	[59,63]
DR5/DRA	A-1331852, BV6	Targeting, accumulation, controlled release	[68]

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
