# Peer review of "Application of Elastin-like Polypeptide in Tumor Therapy"

_cancers, 2022, doi:10.3390/cancers14153683_

Round 1

Reviewer 1 Report

Shi et al reviewed the application of elastin-like polypeptide (ELPs) in tumour therapy. The authors reviewed the ELPs increase the pharmacokinetic properties of therapeutic drugs, ELP fused with specific peptides to improve anti-tumour efficacy of drugs, Conjugation of ELP and peptide/antibody drugs with photosensitizers, and finally Concluded the usefulness of ELPs in tumour therapy. The manuscript is interesting and well written. Still, I believe that the manuscript requires a few revisions before accepting in Cancers.

1. Figures are primitive and required clear planning and adding batter self-explainable figures should be added to the manuscript.

2. Future perspectives missing in this review manuscript.

3. Clinical development of ELPs should be discussed in a separate section. 

Author Response

Thank you for your nice comments on our article. According to your nice suggestions, we have made extensive corrections to our previous draft, the detailed corrections are listed in response file.

Reviewer 2 Report

The article provides valuable insight into high-impact studies on Elastin-Like Polypeptides and their applications in drug delivery and photodynamic therapy. The flow of logic is easy to follow, and information is presented concisely and informative. The article discusses ELP’s effects on increasing the pharmacokinetics of anti-tumour drugs as well as increasing the efficacy of photosensitizers in anti-tumour therapy. The authors identify critical advantages of ELP compared to similar treatments and address their limitations. However, a practical summary of current findings to propose clear areas of future research is necessary. Hence a few suggestions are outlined for the authors’ consideration.

1.     Some more figures are suggested to be included to illustrate the topics clearly.

2.     In the introduction, a brief description of what “various scientific and biomedical applications” are could provide more clarity.

3.     It is helpful to include a few sentences at the end of each section (2-4) to summarize critical information and help the reader better retain the takeaway message.

4.     Does the absence of ELP decrease the accumulation of drugs on the tumour surface in the CPP/TP targeted drug delivery? This information may better clarify the effects of ELP alone and demonstrate the extent of its temperature-sensitive properties in addition to CPP/TP.

5.     Does ELP function most optimally at physiological temperatures? Also, does manipulating temperature possibly affect peptide drug activity?

6.     Perhaps mention if there are known mechanisms of ELP/peptide drug clearance that would also prevent cytotoxicity.

7.     Clarify what “Tt” means in Figure 1. E.g. Indicate increasing temperature above Tt aggregates ELPs.

8.     Perhaps delve more into the limitations for ELP drug delivery as only positives were discussed. Such as obstacles to internalization or decreased efficacy for entering different cancer tissue types.

9.     Mention possible side effects when using ELP in anti-tumour therapy.

10.  The conclusion section should focus on summarizing critical points discussed in the paper. Extra references can be included in a “Future Implications” section to propose the focus of future research.

11.  List specific improvements that could be made to current ELP models to address any weaknesses for future investigations.  

Author Response

We feel great thanks for your professional review work on our article. According to your nice suggestions, we have made extensive corrections to our previous draft, the detailed corrections are listed below.

Comment

  1. Some more figures are suggested to be included to illustrate the topics clearly.

Response

We improve the quality and add new figure to illustrate the key topics clearly.

Comment

  1. In the introduction, a brief description of what “various scientific and biomedical applications” are could provide more clarity.

Response

Further detail on the commonly investigated scientific and biomedical applications of the polypeptides (ELP, RLP and CLP) has been added. ‘The intrinsic property of these polypeptides allows them to fine-tuned into different targeted structures in the form of nanoparticles, nanofibers and other forms of nanocomposites suitable for emerging novel applications of which tissue engineering, drug delivery, bioimaging, biosensor and protein purification are among the most commonly investigated’ (Line 45-49)

Comment

  1. It is helpful to include a few sentences at the end of each section (2-4) to summarize critical information and help the reader better retain the takeaway message.

Response

A summary of the basic idea in each of the sections indicated have been included at the end of each section.

Comment

  1. Does the absence of ELP decrease the accumulation of drugs on the tumour surface in the CPP/TP targeted drug delivery? This information may better clarify the effects of ELP alone and demonstrate the extent of its temperature-sensitive properties in addition to CPP/TP.

Response

This is an interesting point. We found that uptake and accumulation of ELP-drug conjugate may be cell/tissue specific and uses different mechanisms to attach to cell surfaces or penetrate cells. Some studies found that ELP whether alone or in combination with a non-site-specific peptides does not possess the binding required for inducing sustained cytotoxicity. Also, ELP alone is an inert peptide (as known at present) and does not induce immune response. On the other hand, cell penetrating peptide and therapeutic peptides are mostly of low molecular weight and hence have higher clearance rate and low half-life. However, when constructed together, the ELP offers a unique phase transition property including solubility and coagulation to the CPP and TP that may facilitate penetration and stabilization. The major parameter that is required to ensure efficient binding at the tumor site is identifying and employing a specific antibody after which the ELP provides the stability and limits clearance after binding. (Details to this is included at Line 242-250 and Line 390-395)

Comment

  1. Does ELP function most optimally at physiological temperatures? Also, does manipulating temperature possibly affect peptide drug activity?

Response

ELP can remain largely as a soluble macromolecule at physiological temperatures especially when solute concentration is less to induce aggregation. However, after several studies into the uptake of ELP-peptide drugs, Jung Su Ryu has indicated and confirmed that accumulation only occurs when the temperature is ~42°C. At physiological temperatures, however, intravenously delivered ELP-CPP/TP conjugates are quickly cleared from the system, indicating that ELP penetration and aggregation in tissues may require a manipulation in temperature (especially hyperthermia) to achieve a significant degree of cytotoxicity when used for drug delivery. (Details added Line 271-276)

Comment

  1. Perhaps mention if there are known mechanisms of ELP/peptide drug clearance that would also prevent cytotoxicity.

Response

The concept of peptide-based drug clearance has been widely accepted to be achieved through metabolism, degradation, serum and renal clearance. The authors we reviewed so far did not focus much in detail on how the clearance of ELP/PT occurs. However, we postulate based on the knowledge gathered that certain factors must be in place for ELP clearance to occur that will illicit no cytotoxic effect. (Details in Line 212-224)

Comment

  1. Clarify what “Tt” means in Figure 1. E.g. Indicate increasing temperature above Tt aggregates ELPs.

Response

We use “T>Tt” to replace Tt and clarify “T>Tt” means in figure 1.

Comment

  1. Perhaps delve more into the limitations for ELP drug delivery as only positives were discussed. Such as obstacles to internalization or decreased efficacy for entering different cancer tissue types.

Response

A new section ‘‘Limitations and Future perspective for ELP-drug conjugate’ has been added to address this concern (Line 380-445). With respect to ELP-TP conjugate infiltration and internalization, we perceive that the process might be cell/tissue specific and may be dependent on the interaction that occurs between cell surface molecules and the conjugated drug. See details in Line 390-398.

Comment

  1. Mention possible side effects when using ELP in anti-tumour therapy.

Response

Interestingly, almost all the studies we reviewed are of common understanding of the inert nature of ELP and premise their investigation into ELP-peptide drug conjugate on the basis that ELPs are biodegradable, non-toxic biopolymer which does not stimulate any known immune response. As a result, no side effects have been mentioned or reviewed at all at the moment, and we wouldn’t want to thread this fine line safely without speculations.

Comment

  1. The conclusion section should focus on summarizing critical points discussed in the paper. Extra references can be included in a “Future Implications” section to propose the focus of future research.

Response

The conclusion had been revised and restricted to only the critical findings. Other points have been shifted to the newly added section ‘Limitations and Future perspective for ELP-drug conjugate’

Comment

  1. List specific improvements that could be made to current ELP models to address any weaknesses for future investigations.

Response

A single modification to ELP conjugate drug may not provide a one-size fit all solution to all tumors and thus may require trial and error in many cases. However, knowing the behavior of ELPs, it can be tuned to respond to changes in the tumor environment by changing sequence length, guest residue and by linking it to antibodies and peptides with specificity for the target tumor. This concept has been reiterated many times throughout the manuscript as part of the revision and changes made.

Round 2

Reviewer 2 Report

The authors addressed all of my concerns. Thanks.